# Propofol and Sevoflurane Anesthesia in Early Childhood Do Not Influence Seizure Threshold in Adult Rats

**DOI:** 10.3390/ijerph182312367

**Published:** 2021-11-24

**Authors:** Paweł Piwowarczyk, Elżbieta Rypulak, Justyna Sysiak-Sławecka, Dorota Nieoczym, Katarzyna Socała, Aleksandra Wlaź, Piotr Wlaź, Waldemar Turski, Mirosław Czuczwar, Michał Borys

**Affiliations:** 1II Department of Anesthesiology and Intensive Care, Medical University of Lublin, ul. Staszica 16, 20-081 Lublin, Poland; elzbieta.rypulak@umlub.pl (E.R.); justyna.sysiak-slawecka@umlub.pl (J.S.-S.); czuczwarm@gmail.com (M.C.); michalborys@umlub.pl (M.B.); 2Department of Animal Physiology and Pharmacology, Maria Curie-Skłodowska University, 20-033 Lublin, Poland; dorota.nieoczym@mail.umcs.pl (D.N.); katarzyna.socala@mail.umcs.pl (K.S.); piotr.wlaz@mail.umcs.pl (P.W.); 3Department of Diagnostics and Microsurgery of Glaucoma, Medical University of Lublin, 20-079 Lublin, Poland; aleksandra.wlaz@icloud.com; 4Department of Experimental and Clinical Pharmacology, Medical University of Lublin, 20-090 Lublin, Poland; waldemar.turski@umlub.pl

**Keywords:** propofol, sevoflurane, neurodegeneration, synaptogenesis, seizure threshold, epilepsy, rats

## Abstract

Experimental studies have demonstrated that general anesthetics administered during the period of synaptogenesis may induce widespread neurodegeneration, which results in permanent cognitive and behavioral deficits. What remains to be elucidated is the extent of the potential influence of the commonly used hypnotics on comorbidities including epilepsy, which may have resulted from increased neurodegeneration during synaptogenesis. This study aimed to test the hypothesis that neuropathological changes induced by anesthetics during synaptogenesis may lead to changes in the seizure threshold during adulthood. Wistar rat pups were treated with propofol, sevoflurane, or saline on the sixth postnatal day. The long-term effects of prolonged propofol and sevoflurane anesthesia on epileptogenesis were assessed using corneal kindling, pilocarpine-, and pentylenetetrazole-induced seizure models in adult animals. Body weight gain was measured throughout the experiment. No changes in the seizure threshold were observed in the three models. A significant weight gain after exposure to anesthetics during synaptogenesis was observed in the propofol group but not in the sevoflurane group. The results suggest that single prolonged exposure to sevoflurane or propofol during synaptogenesis may have no undesirable effects on epileptogenesis in adulthood.

## 1. Introduction

On a daily basis, propofol and sevoflurane are administered to millions of children who are to undergo surgical procedures [1]. Although pediatric anesthesia is considered safe based on short-term observations, mounting data from animal studies have proven that general anesthetics administered in early life induce widespread apoptotic neurodegeneration in a developing brain [2,3,4,5]. Moreover, one of the major concerns of the clinical and scientific community is that histological changes observed after exposure to anesthetics are associated with long-term cognitive and behavioral deficits observed in preclinical studies [3,6,7].

The majority of general anesthetics act in the central nervous system (CNS) via two mechanisms: agonism on the γ-butyric acid (GABA) receptor or antagonism on the N-methyl-D-aspartate (NMDA) receptor [8]. Propofol and sevoflurane act via GABA agonism. Propofol (2,6 Disopropyl-phenol) is a short-acting intravenous sedative and anesthetic agent [9]. Sevoflurane (1,1,1,3,3,3-Hexafluoro-2-[fluoromethoxy]propane) is an inhalational agent popular in pediatric anesthesia due to its favorable pharmacokinetic properties [10].

The CNS is most vulnerable to the adverse effects of anesthetics during synaptogenesis [2,4,11]. Brain development does not proceed at a uniform pace; rather, it is believed to occur by leaps and bounds during a period called brain growth spurt (BGS), wherein the majority of neurons are created and will synapse with one another. In this process, the equilibrium between glutamatergic and GABA-ergic stimulation is of vital importance [12]. In rats, synaptogenesis occurs between postnatal days 1 (D1) and 14 (D14) [3].

Substances that act on the GABA or NMDA receptor during a BGS could potentially influence the process of synaptogenesis. Prolonged exposure of developing rats to sevoflurane and propofol has been proven to cause neurodegeneration and apoptosis in the brain [4,13]. When given during synaptogenesis, ethanol, a non-specific GABA agonist, impacts not only the neurotransmitter production, but also the receptor density in various brain regions [14,15]. Changes in the hippocampus, which is one of the most epileptogenic regions of the CNS, include loss of pyramidal cells and neuronal remodeling [16,17,18,19]. Interestingly, the histological changes in neural tissue following ethanol administration during a BGS are similar to those observed during epileptogenesis, the underlying pathophysiological process that leads to the development of epilepsy [20,21,22].

Epilepsy is a brain disorder characterized by repeated seizures that present heterogeneous clinical pictures [23]. Epilepsy develops via a process called epileptogenesis, which may be characterized by multiple histological changes in the brain, including neurodegeneration, axonal damage, and activation of glial cells. Most of the reported histological features of the brain may have been observed when the seizure threshold has already decreased, whereas spontaneous seizures have not yet been reported [24,25,26].

Epilepsy remains the most common pediatric neurological disorder, and 1–2% of the population worldwide suffers from this condition [27,28]. It has been postulated that avoidance of the factors that may trigger the initiation of epileptogenic changes in the brain is a key to the successful management of epilepsy [29]. Factors that may predispose an individual to epileptogenesis have been investigated in the preclinical setting by using experimental epilepsy models, such as pilocarpine, pentylenetetrazole, and corneal kindling models [30,31]. The corneal kindling model is believed to be one of the most useful models in epileptogenesis research [32,33]. It has been suggested that the commonly used antiepileptic drugs and ethanol, when administered during BGS, which is a vulnerable period, may accelerate epileptogenesis in experimental epilepsy models [34,35,36]. What remains to be elucidated is the extent of the potential influence of anesthetics commonly used in pediatric settings on comorbidities that may have resulted from increased neurodegeneration during synaptogenesis. Thus, this study aimed to test the hypothesis that prolonged exposure to sevoflurane and propofol during synaptogenesis may lead to changes in seizure threshold during adulthood in three experimental seizure models.

## 2. Methods

All procedures and experiments were approved by the Ethical Committee of the Medical University of Lublin (Approval No. KE-0254/14/2011 issued on 13 May 2011, with Prof. Grażyna Biała as chairperson) and were performed in accordance with the Helsinki Declaration and European Animal Research Association recommendations. This study was conducted in the laboratory of the Department of Experimental and Clinical Pharmacology of the Medical University of Lublin.

### 2.1. Animals

Pregnant Wistar rats were separately housed in cages, given free access to food and water, and reared under a 12 h light–dark cycle (lights on from 07:00 to 19:00). The temperature was maintained at 21 ± 1 °C, and the humidity was set at 30–70%. Each cage contained a different litter. The rats constituting the different groups were earmarked. An equal number of control and experimental rats were taken from the same litters so that each experiment included a littermate control group. Only male rats were used, including 120 pups from 16 litters, given their reportedly higher susceptibility to the deleterious effects of early-life anesthesia compared with females. On D6, the rat pups were randomly segregated into sevoflurane exposure group, propofol exposure group, and control groups. The body weight of the rats was measured on D6, D30, and D60.

### 2.2. Exposition to Sevoflurane

According to protocol described by Tong et al. [13] the rats in the sevoflurane group (30 rats) were placed in a plastic chamber (EZ Anesthesia, Palmer, MA, USA) and exposed to 2.5–3.5% sevoflurane (Abbott Laboratories, Abott Park, IL, USA) for 6 h with oxygen and air mixture (30% oxygen) as a carrier at a gas flow of 2 L/min. During sevoflurane exposure, the rats were placed on a warming device heated to 38 °C (NPS A3; Midea Group Co., Ltd., Foshan, China). When sevoflurane delivery was stopped after 6 h, the rats were observed to see whether they could move freely; then, they were placed back into the maternal cage. Thirty rats in the control group were placed on the same heater and into the same container as those in the sevoflurane group but were exposed to air for 6 h.

### 2.3. Exposure to Propofol

According to the protocol described by Bercker et al. [4], propofol was injected intraperitoneally to 30 rats at 30 mg/kg body weight (Diprivan, AstraZeneca, GB) every 90 min until a cumulative dose of 90 mg/kg was reached. In the control group, 0.9% NaCl was injected intraperitoneally to 30 rats under the same dosing regimen. The rats in both groups were also placed on a warming device heated to 38 °C. During anesthesia administration in the treatment and control groups, the rats’ skin color, capillary refill time, and respiratory rate were monitored. If signs of apnea or hypoxia were observed, the rats were exposed to air immediately and then excluded from the experiment.

### 2.4. Experimental Seizure Models

We used the pilocarpine model of temporal lobe epilepsy (TLE) according to the method described by Turski et al. [30]. At D60, 31 rats were initially subcutaneously injected with scopolamine methyl bromide (2.5 mg/kg, s.c., Sigma, St. Louis, MI, USA); after 30 min, they were injected with different concentrations of intraperitoneal pilocarpine hydrochloride (5 mL/kg i.p.; Sigma, St. Louis, MI, USA). We aimed to establish the CD50—the dose (mg/kg) necessary to initiate grades 4 and 5 seizures in 50% of rats in the four study groups (sevoflurane group, control group for sevoflurane, propofol group, control group for propofol) within 1 h. Scopolamine methyl bromide was used to prevent the peripheral effects of pilocarpine.

In the second seizure threshold model, we used pentylenetetrazole (Sigma, St. Louis, MI, USA), a proconvulsant that could induce myoclonic seizures [37]. Different pentylene-tetrazole concentrations were injected intraperitoneally to 32 rats in the four study groups with the aim of establishing the CD50. Seizure severity following pentylene-tetrazole administration was evaluated as in the pilocarpine seizure model.

The third seizure threshold model involves corneal kindling, which induces complex partial seizures [38]. To induce seizures, we used copper electrodes covered with a soft material and flushed with 0.9% NaCl and 1% topical lidocaine for corneal protection and analgesic purposes. Subsequently, 8 mA current was applied for 4 s to the cornea of 44 rats in the four study groups. The rats were stimulated twice daily for five days a week according to the scheme presented by Wlaź et al. After equivalent, subliminal, and repeated stimuli were applied, increased vulnerability to subsequent corneal stimulation was achieved. We measured the number of stimulations needed to achieve grades 4 and 5 seizures.

Seizure severity was assessed using the five-grade Racine scale: grade 1—mouth and facial movement; grade 2—head nodding; grade 3—forelimb clonus; grade 4—rearing with forelimb clonus; and grade 5—rearing and falling with forelimb clonus [39].

### 2.5. Statistical Analysis

All statistical calculations were performed using MS Excel (Microsoft, Redmond, WA, USA) and STATISTICA v. 13.3 (StatSoft Inc., Tulsa, OK, USA). A *p*-value of 0.05 or lower was considered statistically significant. Body weight measured on D6, D30, and D60 were expressed as means ± standard deviation. In the corneal kindling model, results are presented as the number of stimulations needed to achieve grades 4 and 5 seizures. We checked the distribution of data for normality with the Shapiro-Wilk test. For multiple comparisons regarding the body weight at D6, D30, D60 we used One-way ANOVA with post hoc Bonferroni test and, for comparison with the control group in the corneal kindling model, the Mann-Whitney test. To calculate the clonus dose 50 (CD50) with 95% confidence intervals after pilocarpine and pentylenetetrazole administration and to make comparisons with the control group, we used log-probit analysis according to Litchfield and Wilcoxon [40].

## 3. Results

### 3.1. Corneal Kindling

Exposure of 6-day old rats to anesthetics sevoflurane and propofol did not result in a significant decrease in the number of stimulations needed to achieve grade 4 and grade 5 seizures in corneal kindling performed on adult animals (Figure 1).

There was no significant difference between the control group for sevoflurane and the sevoflurane group in number of stimulations needed to achieve 4th- (*p* = 0.51) and 5th-grade seizures (*p* = 0.43). Similarly, we did not observe difference between the control group for propofol and the propofol group in the number of stimulations needed to achieve 4th- (*p* = 0.1) and 5th-grade seizures (*p* = 0.24) in the corneal kindling model on D60.

### 3.2. Drug Induced Seizures

We performed additional analyses of the influence of sevoflurane and propofol exposure during synaptogenesis on seizure threshold in adulthood. We found that neither anesthetic affected the CD50 in the pilocarpine (Table 1) and pentylenetetrazole seizure models (Table 2).

Table 1 presents pilocarpine doses (mg/kg), which cause 5th-grade seizures according to the Racine scale in 50% of the studied animals (CD_50_) with 95% confidence intervals. Sevoflurane—CD_50_ in the group of rats exposed to sevoflurane on 6th postnatal day at the concentration of 2.5–3.5 vol % for 6 h; propofol—CD_50_ in the group of rats exposed to propofol on 6th postnatal day (cumulative dose 90 mg/kg during 4.5 h); control—CD_50_ in the control group. *n*—number of rats in the group. Log-probit analysis, according to Litchfield and Wilcoxon, was used. *p* < 0.05 was assumed as statistically significant.

Table 2 presents pentylenetetrazole doses (mg/kg), which cause 5th-grade seizures according to the Racine scale in 50% of the studied animals (CD_50_) with 95% confidence intervals. Sevoflurane—CD_50_ in the group of rats exposed to sevoflurane on 6th postnatal day at the concentration of 2.5–3.5 vol % for 6 h; propofol—CD_50_ in the group of rats exposed to propofol on 6th postnatal day (cumulative dose 90 mg/kg during 4.5 h); control—CD_50_ in the control group. *n*—number of rats in the group. Log-probit analysis, according to Litchfield and Wilcoxon, was used. *p* < 0.05 was assumed as statistically significant.

### 3.3. Body Weight

We measured body weight change throughout the experiment in the studied groups (Figure 2). Body weight measured on the day of anesthesia administration at D6 did not significantly differ among the study groups (*p* = 0.811). Significant difference was observed between the propofol group and propofol CON group on D60 (*p* = 0.004).

Five study animals died during experiment (one in the propofol group and four in the sevoflurane group) mostly in the peri-anesthetic period.

## 4. Discussion

In order to test the hypothesis that neuropathological alterations induced by propofol and sevoflurane during synaptogenesis in rats may lead to changes in seizure threshold during adulthood, we adopted protocols described by Bercker et al. and Tong et al. [4,13].

In his original study, Bercker et al. demonstrated that propofol administered to 6-days old rats at the cumulative dose of 90 mg/kg produced profound neurodegenerative effect. The significantly higher scores of degenerated neurons compared to controls were detected using histological silver nitrate and cupric nitrate staining. Moreover, morphological changes were followed by persistent cognitive and learning deficits [4]. Importantly, neurodegeneration was mainly present in the thalamus and subiculum, which is a part of the hippocampus, one of the most epileptogenic regions of the brain [41]. This protocol was accurately reproduced in our study.

The exposure of rats to sevoflurane was carried out according to Tong et al. [13] with minor modification. In our study rats were younger (D6 vs. D21) and time of exposure was longer (6 h vs. 4 h). Tong et al. performed in-depth morphological analysis of the brain. They have proven that sevoflurane anesthesia in juvenile rats has caused a significant increase in apoptosis in newly-born granule cells of dentate gyrus in rats 2 h, 24 h, 4 days, and 7 days after exposure. Additionally, they demonstrated that exposure to sevoflurane decreased the differentiation of the BrdU-labeled late-stage progenitor granule cells but increased the expression of caspase-3, autophagy, and phosphorylated-P65 in the hippocampus of juvenile rats and resulted in cognitive deficiency [13]. Noteworthy, dentate gurus, which is a region of the hippocampus, is strictly associated with epileptogenesis [42].

Furthermore, there are many in vivo and in vitro reports that confirm neurodegeneration after anesthetic regimen similar to the one used in the presented manuscript. According to Fang et al. 4-h sevoflurane exposure of rats on D7 decreased the number of BrdU-positive cells in the hippocampal dentate gyrus and Ki-67 mRNA, indicating that neurogenesis is suppressed. Sevoflurane additionally enhanced Fluoro-Jade staining in the dentate gyrus and induced caspase-3 activation indicative of neuronal cell death until at least 4 days after anesthesia. Spatial reference memory was affected at 6 weeks after sevoflurane administration [43]. Moreover, Zhao et al. has confirmed that 3% sevoflurane administered for 2 h daily from postnatal day 6 to P8 induced neuronal apoptosis and neurogenesis inhibition through the PPARγ signaling pathway in 6-day-old mice, which impaired learning and memory in young mice [44].

Our results demonstrate that long-lasting exposure of rats to propofol and sevoflurane during synaptogenesis affected neither the seizure threshold in the pilocarpine and pentylenetetrazole seizure threshold models nor the number of stimulations needed to achieve grades 4 and 5 seizure in the corneal kindling model. Additionally, we found that propofol administration during BGS caused significant weight gain in the treatment group during the follow-up period (D60) relative to the control group.

The initial phase of the clinical picture of epileptogenesis involves a decrease in seizure threshold, followed by the occurrence of spontaneous seizures; the final phase involves an increase in the intensity and frequency of epileptic seizures [45]. The seizure threshold in the groups exposed to sevoflurane and propofol administered according to the regime that causes neurodegeneration and in the control group was assessed on D60 in the three experimental seizure models. The study design reported herein evaluated vulnerability to epileptogenesis of rat brains in adulthood. The corneal kindling model used in this study is believed to be one of the most useful models in epileptogenesis research [32,33,34]. We have not observed a decrease in seizure threshold in any of the seizure threshold models. Contrary to our results, the findings of Tagashira et al. have shown a decrease in seizure threshold in adult rats in response to the administration of substances that act as GABA receptor agonists during a BGS [46]. In the aforementioned study, ethanol (17.8 g/kg), phenobarbital (55.3 g/kg), diazepam (128.8 g/kg), and barbital (258 g/kg) administered daily from D3 to D21 decreased the seizure threshold after 42 days in all three study groups in the pentylenetetrazole model. The influence of ethanol administration during synaptogenesis on the risk of epileptogenesis in adulthood has been postulated by many authors [47,48]. Bonthius et al. investigated the histological changes in the hippocampus and assessed the seizure threshold in rats after a repetitive, 5-day ethanol administration during BGS in two study groups (2.5 and 3.75 g/kg/day) and after a single dose of 3.75 g/kg ethanol in a third group [36]. On D90, the loss of CA1 pyramidal cells in the hippocampus was significantly more pronounced in the group subjected to a 5-day, repetitive 3.75 g/kg/day ethanol administration. Moreover, the seizure threshold in the pentylenetetrazole seizure threshold model decreased proportionally to the loss of pyramidal cells. Electrophysiological analysis of seizures has confirmed that the main epileptogenic structure responsible for the initiation of seizures is the hippocampus. A single exposure to ethanol neither caused histological changes nor decreased the seizure threshold in the pentylenetetrazole seizure threshold model [36]. It is worth noting that epileptogenesis occurred only after repetitive exposure to ethanol and was proportional to the degree of hippocampus damage. Moreover, Kang et al. have observed a decrease in GABA receptors on the CA1 hippocampal cells following a multiple-dose ethanol treatment during BGS [49].

To exclude the influence of sedation after general anesthesia on the rats’ capability to obtain food and water, which could influence growth, rats’ body weight was monitored. No differences were observed until D60. The significant weight gain was found on D60 in the group exposed to propofol but not sevoflurane. Bercker et al. have not observed differences in body weight 24 h after an equivalent anesthetic exposure (90 mg/kg of cumulative dose of propofol and 3–5 vol. % sevoflurane for 6 h) [4]. Similarly, Makaryus et al. have reported that administration of 2.2 vol % sevoflurane for 5 h on D7 and D15 did not influence body weight 48 h after the termination of anesthesia [50]. As regards the other substances that act as GABA agonists, Peraino et al. have proven that early life exposure to phenobarbital causes a significant decrease in body weight after three days, whereas Bonthius et al. did not observe any influence on weight gain of single doses of ethanol (0.85, 2.75, and 3.75 g/kg) given enterally during synaptogenesis [36,51]. The significant weight gain after propofol administration during BGS found in our study may be attributed to the influence of the lipid solutions present in the pharmaceutical formula of propofol. However, further investigations are needed to explain this finding.

Our study has many limitations. Following the regulations of the Helsinki declaration minimizing the number of subjects in the experiment, we did not perform a histological examination in the study groups after anesthesia. The occurrence of neurodegeneration after exposure to sevoflurane and propofol was based on the results obtained by Bercker et al., Tong et al., Zhao et al., and Satomoto et al. [4,13,44,52]. We checked the seizure thresholds only in three experimental epilepsy models, and the influence of anesthesia during BGS on seizure threshold in other models may differ from our results. Moreover, we did not investigate the influence of prolonged or repetitive anesthetic exposure on seizure threshold.

## 5. Conclusions

Our results suggest that single exposure to sevoflurane or propofol for 6 h during synaptogenesis does not affect the seizure threshold in adult rats. Propofol use during synaptogenesis in rats may be associated with weight gain. Further studies are needed to elucidate the mechanism responsible for our findings.

## Figures and Tables

**Figure 1 ijerph-18-12367-f001:**
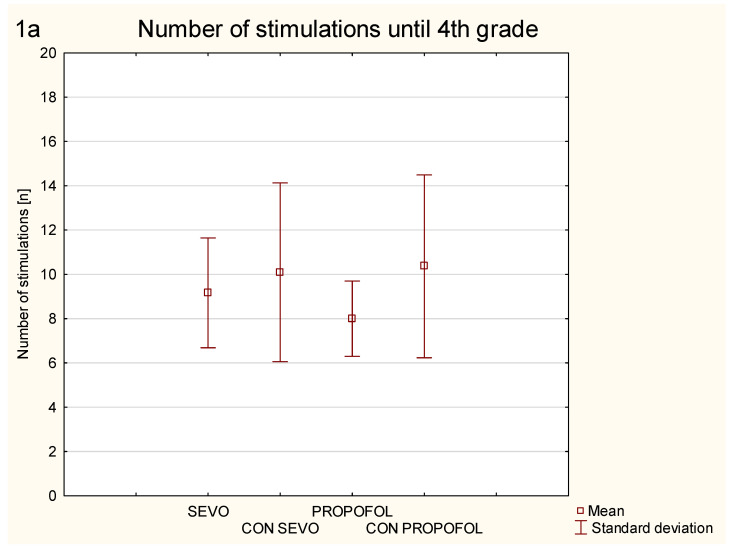
Presents the number of stimulations needed to achieve 4th grade seizures (**1a**) and 5th grade seizures (**1b**) according to the Racine scale in corneal kindling model: SEVO—mean number of stimulations in the group of rats exposed to sevoflurane on 6th postnatal day at the concentration of 2.5–3.5 vol % for 6 h; PROPOFOL—mean number of stimulations in the group of rats exposed to propofol (cumulative dose 90 mg/kg during 4.5 h); CON SEVO—mean number of stimulations in the control group for sevoflurane; CON PROPOFOL—mean number of stimulations in the control group for propofol. Whiskers represent standard deviation. Mann-Whitney test was used. *p* < 0.05 was assumed as statistically significant.

**Figure 2 ijerph-18-12367-f002:**
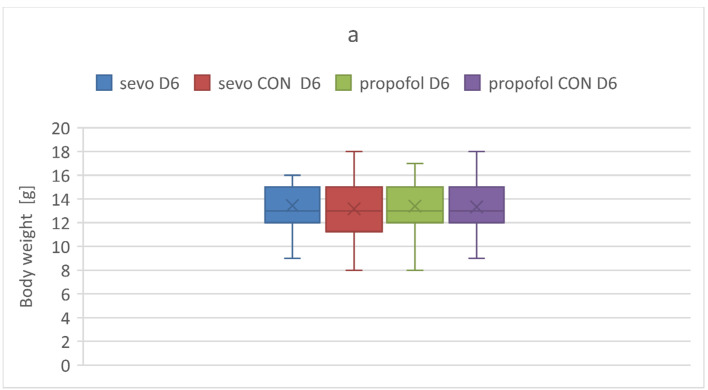
Rats’ body weight at D6, D30, D 60. (**a**) presents mean body weight [g] in the studied groups at D6. (**b**) presents mean body weight [g] in the studied groups at D30. (**c**) presents mean body weight [g] in the studied groups at D60: sevo D6; sevo D30; sevo D60—mean body weight in the group randomized to sevoflurane at D6, D30, D60; propofol D6; propofol D30; propofol D60—mean body weight in the group randomized to propofol at D6, D30, D60; sevo CON D6; sevo CON D30; sevo CON D60- mean body weight in the group randomized to control for sevoflurane at D6, D30, D60; propofol CON D6; propofol CON D30; propofol CON D60—mean body weight in the groups randomized to control for propofol at D6, D30, D60. Crosses represent mean. Whiskers represent standard deviation. Dots represent outliers. Boxes represent inter-quartile range. Horizontal lines represent medians. Data were checked for normality with the Shapiro-Wilk test. One-way ANOVA was used for multiple comparison with post hoc Bonferroni test. * *p* < 0.05 was assumed as statistically significant.

**Table 1 ijerph-18-12367-t001:** Influence of propofol and sevoflurane administration during synaptogenesis on seizure threshold in pilocarpine model in adult rats.

Substance	CD_50_ Pilocarpine (mg/kg)	*n*	*p*
Propofol control group	344.1 (337.7–350.7)	7	0.63
Propofol	339.4 (330–349.1)	8	
Sevoflurane control group	341.9 (317–368.7)	8	0.42
Sevoflurane	323.3 (305–362.1)	8	

**Table 2 ijerph-18-12367-t002:** Influence of propofol and sevoflurane administration during synaptogenesis on seizure threshold in pentylene-tetrazole model in adult rats.

Substance	CD_50_ Pentylenetetrazole (mg/kg)	*n*	*p*
Propofol control group	108.5 (92.8–126.8)	8	0.21
Propofol	95.6 (77.9–117.3)	8	
Sevoflurane control group	104.9 (86.8–126.9)	8	0.19
Sevoflurane	94.3 (78–114)	8	

## Data Availability

The datasets used and/or analyzed in the current study are available from the corresponding author upon reasonable request.

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
