# Peer review of "Propofol and Sevoflurane Anesthesia in Early Childhood Do Not Influence Seizure Threshold in Adult Rats"

_ijerph, 2021, doi:10.3390/ijerph182312367_

Round 1
Reviewer 1 Report
This is an interesting research about the impact of anesthetics in seizure threshold during synaptogenesis. Therefore, an important concern about this study is that the authors proposed “to test the hypothesis that apoptosis induced by sevofluorane and propofol during synaptogenesis may lead to changes in seizure threshold…”. After a careful reading of the manuscript, I was unable to identify any evidence about apoptosis in the experiment. My suggestion is for the authors to describe which technique and results they obtained inducing apoptosis by the anesthetics employed in the research.
The manuscript strengths are: first, the introduction is appropriate, describing the problem of seizures and how the model employed in the experiment is useful in epilepsy research; second, the animal model was appropriate for the proposed research; third, statistical calculations are adequate. Otherwise, this paper has an important weakness. The main hypothesis, “that apoptosis induced by sevofluorane and propofol during synaptogenesis may lead to changes in seizure threshold during adulthood…” is not proved in the results. The authors did not describe how they induced or proved apoptosis in the experiment. So, it’s very hard to establish a cause-effect phenom between the anesthetics employed, the experimental seizure model and to answer the research hypothesis.
To improve the manuscript the authors should:
- To describe the methodology employed to provoke and show apoptosis in this model.
- Starting discussion describing the association between the level of apoptosis needed to promote changes in seizure threshold in adulthood in the animals submitted to exposure to sevofluorane and propofol in synaptogenesis period.
- If this data is unavailable, the experiment hypothesis should be modified.
Author Response
Thank you for your comments and suggestions. I have addressed all points highlighted in both reviews (see below). To improve the readability of the corrections in the manuscript color marking was used.
Reviewer 1
Comments and Suggestions
- This is an interesting research about the impact of anesthetics in seizure threshold during synaptogenesis. Therefore, an important concern about this study is that the authors proposed “to test the hypothesis that apoptosis induced by sevofluorane and propofol during synaptogenesis may lead to changes in seizure threshold…”. After a careful reading of the manuscript, I was unable to identify any evidence about apoptosis in the experiment. My suggestion is for the authors to describe which technique and results they obtained inducing apoptosis by the anesthetics employed in the research.
Authors’ response:
The anesthetic protocol regarding propofol anesthesia applied in the presented manuscript was identical as in the study presented by Bercker et al. (2009). The authors of the aforementioned study have demonstrated that anesthesia with a cumulative intraperitoneal dose of 90 mg/kg of propofol caused significantly higher scores of degenerated neurons compared to controls in histological silver nitrate and cupric nitrate staining of rats anesthetized on sixth postnatal day of life. Neurodegenerative changes were mainly present in the thalamus and in subiculus, which is a part of the hippocampus, one of the most epileptogenic regions of the brain. According to the study by Bercker et al. no changes in the CA1 and dentate gyrus were observed after administration of propofol [4,41].
The anesthetic protocol regarding sevoflurane anesthesia in the presented study was based on 6 hour exposure to sevoflurane at the concentrations of 2,5-3,5 vol% on the sixth postnatal day. Using a similar protocol, Tong et al. have proven that 4-hour anesthesia with sevoflurane at the concentration of 3 vol% has caused a significant increase in apoptosis in newly-born granule cells of dentate gyrus in rats 2 h, 24 h, 4 days, and 7 days after exposure in juvenile rats. Denate gurus, which is a region of the hippocampus, is strictly associated with epileptogenesis [13,42].
According to the regulations of the Helsinki Declaration, to minimize the number of animals in the experiments that have already been conducted, we did not repeat histological examination.
- Bercker S, Bert B, Bittigau P i wsp.. Neurodegeneration in newborn rats following propofol and sevoflurane anesthesia. Neurotox Res. 2009; 16: 140-147.
- Lévesque M, Avoli M. The subiculum and its role in focal epileptic disorders. Rev Neurosci. 2020 Nov 30;32(3):249-273. doi: 10.1515/revneuro-2020-0091. PMID: 33661586.
- Tong D, Ma Z, Su P, Wang S, Xu Y, Zhang LM, Wu Z, Liu K, Zhao P. Sevoflurane-Induced Neuroapoptosis in Rat Dentate Gyrus Is Activated by Autophagy Through NF-κB Signaling on the Late-Stage Progenitor Granule Cells. Front Cell Neurosci. 2020 Dec 15;14:590577.
- Dengler CG, Coulter DA. Normal and epilepsy-associated pathologic function of the dentate gyrus. Prog Brain Res. 2016;226:155-178. doi:10.1016/bs.pbr.2016.04.005
- Fang F, Xue Z, Cang J. Sevoflurane exposure in 7-day-old rats affects neurogenesis, neurodegeneration and neurocognitive function. Neurosci Bull. 2012 Oct;28(5):499-508.
- Zhao Y, Chen K, Shen X. Environmental Enrichment Attenuated Sevoflurane-Induced Neurotoxicity through the PPAR-γ Signaling Pathway. Biomed Res Int. 2015;2015:107149.
We have included relevant additional sentences in the introduction, methods section, and discussion.
In the Introduction:
„Prolonged exposure of developing rats to sevoflurane and propofol has been proven to cause neurodegeneration and apoptosis in the brains [4,13]”
In the Methods:
„According to protocol described by Tong et al. (2020) [13] the rats in the sevoflurane group (thirty rats) were placed in a plastic chamber (EZ Anesthesia, Palmer, USA) and exposed to 2.5–3.5% sevoflurane (Abbott Laboratories, Abott Park, IL, USA) for 6 h with oxygen and air mixture (30% oxygen) as a carrier at a gas flow of 2 l/min.”
„According to the protocol described by Bercker et al. (2009) [4] propofol was injected intraperitoneally to thirty rats at 30 mg/kg body weight (Diprivan, AstraZeneca, GB) every 90 min until a cumulative dose of 90 mg/kg was reached. In the control group, 0.9% NaCl was injected intraperitoneally to thirty rats under the same dosing regimen.”
In the Discussion:
In order to test the hypothesis that neuropathological alterations induced by propofol and sevoflurane during synaptogenesis in rats may lead to changes in seizure threshold during adulthood we adopted protocols described by Bercker et al. and Tong et al. [4,13].
In his original study Bercker et al. has demonstrated that propofol administered to 6-days old rats at the cumulative dose of 90 mg/kg produced profound neurodegenerative effect. The significantly higher scores of degenerated neurons compared to controls were detected using histological silver nitrate and cupric nitrate staining. Moreover, morphological changes were followed by persistent cognitive and learning deficits [4]. Importantly, neurodegeneration was mainly present in the thalamus and in subiculus, which is a part of the hippocampus, one of the most epileptogenic regions of the brain [41]. This protocol was accurately reproduced in our study.
In our study the exposure of rats to sevoflurane was carried out according to Tong et al. [13] with minor modification. In our study rats were younger (D6 vs D21) and time of exposure was longer (6 h vs 4 h). Tong and co-workers performed in-depth morphological analysis of the brain. They have proven that sevoflurane anesthesia in juvenile rats has caused a significant increase in apoptosis in newly-born granule cells of dentate gyrus in rats 2 h, 24 h, 4 days, and 7 days after exposure. Additionally, they demonstrated that exposure to sevoflurane decreased the differentiation of the BrdU-labeled late-stage progenitor granule cells but increased the expression of caspase-3, autophagy, and phosphorylated-P65 in the hippocampus of juvenile rats and resulted in cognitive deficiency [13]. Noteworthy, denate gurus, which is a region of the hippocampus, is strictly associated with epileptogenesis [42].
Furthermore, there are many in vivo and in vitro reports that confirm neurodegeneration after anesthetic regimen similar to the one used in the presented manuscript. According to Fang et al. 4-hour sevoflurane exposure on D7 decreased the number of BrdU-positive cells in the hippocampal dentate gyrus and Ki-67 mRNA, indicating neurogenesis is suppressed. Sevoflurane additionally enhanced Fluoro-Jade staining in the dentate gyrus and induced caspase-3 activation indicative of neuronal cell death until at least 4 days after anesthesia. Spatial reference memory was affected at 6 weeks after sevoflurane administration [43]. Moreover, Zhao et al. has confirmed that 3% sevoflurane administered for 2 hours daily from postnatal day 6 to P8 induced neuronal apoptosis and neurogenesis inhibition through the PPAR? signaling pathway in 6-day-old mice, which impaired learning and memory in young mice [44].”
The manuscript strengths are: first, the introduction is appropriate, describing the problem of seizures and how the model employed in the experiment is useful in epilepsy research; second, the animal model was appropriate for the proposed research; third, statistical calculations are adequate.
- Otherwise, this paper has an important weakness. The main hypothesis, “that apoptosis induced by sevofluorane and propofol during synaptogenesis may lead to changes in seizure threshold during adulthood…” is not proved in the results. The authors did not describe how they induced or proved apoptosis in the experiment. So, it’s very hard to establish a cause-effect phenom between the anesthetics employed, the experimental seizure model and to answer the research hypothesis.
Authors’ response: Thank you for this remark. We have enlisted the evidence of neurodegeneration after the anesthetic regimen applied in the presented study and how the used protocol addressed the regulation of the Helsinki Declaration in response to the first question. We have enlisted references to the relevant literature (see repley above).
To improve the manuscript the authors should:
- To describe the methodology employed to provoke and show apoptosis in this model.
Authors’ response: Thank you for this remark. We have enlisted the evidence of neurodegeneration after the anesthetic regimen applied in the presented study and how the used protocol addressed the regulation of the Helsinki Declaration in response to the first question. We have enlisted references to the relevant literature (see repley above).
- Starting discussion describing the association between the level of apoptosis needed to promote changes in seizure threshold in adulthood in the animals submitted to exposure to sevofluorane and propofol in synaptogenesis period.
Authors’ response:
Thank you for this interesting remark. The level of apoptosis needed to promote changes in the seizure threshold, especially after sevoflurane and propofol anesthesia, has not yet been established. There are no data regarding this issue in the literature.
There are many studies that confirm apoptosis after the use of multiple anesthetics: e.g., propofol, isoflurane, sevoflurane used at different treatment regimens in different species and those that confirm synergy in neurodegenerative properties of their combinations: e.g., midazolam + isoflurane [1].
To the best of our knowledge, the presented study is the first that tested if there is an association between anesthetics given during synaptogenesis and decreased seizure threshold in adulthood.
There are studies describing histological changes present during epileptogenesis, but still, quantification and association of those changes with decreased seizure threshold have not yet been done.
There are, however neurodegenerative diseases, in which the level of apoptosis needed to promote symptoms is established. According to the literature, symptoms in Parkinson’s disease occur only when more than 80% of dopaminergic neurons at substantia nigra are damaged [2].
It would be very promising to establish an association between the level of histological damage and the presence of symptoms in epilepsy. This issue may be a subject of future research.
Instead, at the begining of the Discussion section neuropathological changes induced by propofol and sevoflurane during synaptogenesis in rats were extensively described.
- Jevtovic-Todorovic V, Hartman RE, Izumi Y, Benshoff ND, Dikranian K, Zorumski CF, Olney JW, Wozniak DF. Early exposure to common anesthetic agents causes widespread neurodegeneration in the developing rat brain and persistent learning deficits. J Neurosci. 2003;23(3):876-82.
- Dauer W, Przedborski S. Parkinson's disease: mechanisms and models. Neuron. 2003;39(6):889-909. doi: 10.1016/s0896-6273(03)00568-3. PMID: 12971891.
- If this data is unavailable, the experiment hypothesis should be modified
Authors’ response: Thank you for this remark. Accordingly, the aim of the study was reedited as follows: „Thus, this study aimed to test the hypothesis that prolonged exposure to sevoflurane and propofol during synaptogenesis may lead to changes in seizure threshold during adulthood in three experimental seizure models.”
Moreover, the evidence of neurodegeneration that is provoked by the anesthetic regimen applied in the present study was extensively described in Discussion (see repley above) and relevant literature references were added.
Reviewer 2
Dear reviewer,
thank you for Your comments and suggestions. I have addressed all points highlighted in Your review below and used colour marking in the manuscript to improve readability of the corrections.
Piwowarczyk et al., showed that childhood exposure to propofol and sevoflurane anesthesia did not affect the seizure threshold in adult rats. It is an interesting piece of study having a translational value. It is a nicely written manuscript, easy to follow. However, I have few suggestions/ questions to improve the quality and readability of the article.
- Abstract: any kind of numerical values are unnecessary in the abstract.
Authors’ response: We have removed numerical values from the abstract according to your suggestions.
- Have you tried NMDAR based anesthesia, if not why only GABA based?
Authors’ response: We have not tried to investigate the influence of agents that work via NMDAR antagonism on seizure threshold. We focused on the two most commonly used agents in the pediatric setting: propofol and sevoflurane. Use of NMDAR antagonists during synaptogenesis will be our next focus of research.
- Does the dose and duration of the anesthesia sufficient to induce synaptogenesis. Need to be mentioned clearly.
Authors’ response: The anesthetic protocol regarding propofol anesthesia applied in the presented manuscript was identical as in the study presented by Bercker et al. (2009). The authors of the aforementioned study have demonstrated that anesthesia with a cumulative intraperitoneal dose of 90 mg/kg of propofol caused significantly higher scores of degenerated neurons compared to controls in histological silver nitrate and cupric nitrate staining of rats anesthetized on sixth postnatal day of life. Neurodegenerative changes were mainly present in the thalamus and in subiculus, which is a part of the hippocampus, one of the most epileptogenic regions of the brain. According to the study by Bercker et al. no changes in the CA1 and dentate gyrus were observed after administration of propofol [1,2].
The anesthetic protocol regarding sevoflurane anesthesia in the presented study was based on 6 hour exposure to sevoflurane at the concentrations of 2,5-3,5 vol% on the sixth postnatal day. Using a similar protocol, Tong et al. have proven that 4-hour anesthesia with sevoflurane at the concentration of 3 vol% has caused a significant increase in apoptosis in newly-born granule cells of dentate gyrus in rats 2 h, 24 h, 4 days, and 7 days after exposure in juvenile rats. Denate gurus, which is a region of the hippocampus, is strictly associated with epileptogenesis [3,4].
According to the regulations of the Helsinki Declaration, to minimize the number of animals in the experiments that have already been conducted, we did not repeat histological examination.
We have included relevant additional sentences in the introduction, methods section, and discussion.
In the Introduction:
„Prolonged exposure of developing rats to sevoflurane and propofol has been proven to cause neurodegeneration and apoptosis in the brains [4,13]”
In the Methods:
„According to protocol described by Tong et al. (2020) [13] the rats in the sevoflurane group (thirty rats) were placed in a plastic chamber (EZ Anesthesia, Palmer, USA) and exposed to 2.5–3.5% sevoflurane (Abbott Laboratories, Abott Park, IL, USA) for 6 h with oxygen and air mixture (30% oxygen) as a carrier at a gas flow of 2 l/min.”
„According to the protocol described by Bercker et al. (2009) [4] propofol was injected intraperitoneally to thirty rats at 30 mg/kg body weight (Diprivan, AstraZeneca, GB) every 90 min until a cumulative dose of 90 mg/kg was reached. In the control group, 0.9% NaCl was injected intraperitoneally to thirty rats under the same dosing regimen.”
In the Discussion:
In order to test the hypothesis that neuropathological alterations induced by propofol and sevoflurane during synaptogenesis in rats may lead to changes in seizure threshold during adulthood we adopted protocols described by Bercker et al. and Tong et al. [13].
In his original study Bercker et al. has demonstrated that propofol administered to 6-days old rats at the cumulative dose of 90 mg/kg produced profound neurodegenerative effect. The significantly higher scores of degenerated neurons compared to controls were detected using histological silver nitrate and cupric nitrate staining. Moreover, morphological changes were followed by persistent cognitive and learning deficits [1]. Importantly, neurodegeneration was mainly present in the thalamus and in subiculus, which is a part of the hippocampus, one of the most epileptogenic regions of the brain [2]. This protocol was accurately reproduced in our study.
In our study the exposure of rats to sevoflurane was carried out according to Tong et al. [13] with minor modification. In our study rats were younger (D6 vs D21) and time of exposure was longer (6 h vs 4 h). Tong and co-workers performed in-depth morphological analysis of the brain. They have proven that sevoflurane anesthesia in juvenile rats has caused a significant increase in apoptosis in newly-born granule cells of dentate gyrus in rats 2 h, 24 h, 4 days, and 7 days after exposure. Additionally, they demonstrated that exposure to sevoflurane decreased the differentiation of the BrdU-labeled late-stage progenitor granule cells but increased the expression of caspase-3, autophagy, and phosphorylated-P65 in the hippocampus of juvenile rats and resulted in cognitive deficiency [3]. Noteworthy, denate gurus, which is a region of the hippocampus, is strictly associated with epileptogenesis [4].
Furthermore, there are many in vivo and in vitro reports that confirm neurodegeneration after anesthetic regimen similar to the one used in the presented manuscript. According to Fang et al. 4-hour sevoflurane exposure on D7 decreased the number of BrdU-positive cells in the hippocampal dentate gyrus and Ki-67 mRNA, indicating neurogenesis is suppressed. Sevoflurane additionally enhanced Fluoro-Jade staining in the dentate gyrus and induced caspase-3 activation indicative of neuronal cell death until at least 4 days after anesthesia. Spatial reference memory was affected at 6 weeks after sevoflurane administration [5]. Moreover, Zhao et al. has confirmed that 3% sevoflurane administered for 2 hours daily from postnatal day 6 to P8 induced neuronal apoptosis and neurogenesis inhibition through the PPAR? signaling pathway in 6-day-old mice, which impaired learning and memory in young mice [6].”
- Introduction is too long; it can be short and concise.
Authors’ response: We have shortened the introduction according to your suggestion. The paragraph about epilepsy and a few sentences in the paragraph that described anesthetics have been removed. (overall reduction by 112 words.)
- Results: Most of the results written in the text are duplicated in the figures. Values written in the text can be removed to improve the readability. However, these values can be added to the graphs if needed.
Authors’ response: We have removed the duplicated data and numerical values from the text according to your suggestions to improve the readability.
- Different color for different treatments would improve the figures.
Authors’ response: We have used a different color for the three different study groups according to your suggestions.
- For comparisons of the multiple groups, One-Way-ANOVA, which is a more powerful text, should be used.
Authors’ response: Thank you for your remark. We have compared multiple groups regarding body weight with the use of One Way ANOVA instead of comparing only the study group with the control group with the Mann-Whitney test. In regards to the corneal kindling model - due to the fact that this seizure model is based on ordinal values instead of continuous values, we used the Mann-Whitney U Test.
- All graphs should be combined into one or max two figures. The figure can have sub-figures.
Authors’ response: We have changed the number of figures from 5 to 2. The first figure depicts the influence of anesthetics on 4th and 5th grade seizures in the corneal kindling model, and the second figure depicts rats' body weight at D6, D30, and D60 in the four different study groups.
- Tables: There is confusion of “coma (,)” and “decimal (.)” in the values. As a standard usages decimals are used as “.” Not coma.
Authors’ response: We have changed coma (,) into decimal (.) in all the values in the manuscript according to your suggestion.
- I think the table-1 and table-2 are exchanged; the description looks mismatched.
Authors’ response: There was no separation between table 2 and the manuscript paragraph about body weight. We have increased the space between table 2 and the manuscript not to confuse the reader.
- Why term “body mass” is used instead of body weight, explain. If no specific purpose, then “body weight” is standard usages.
Authors’ response: We have changed the term „ body mass” to „body weight” in the manuscript according to your suggestions.

Reviewer 2 Report
Piwowarczyk et al., showed that childhood exposure to propofol and sevoflurane anesthesia did not affect the seizure threshold in adult rats. It is an interesting piece of study having a translational value. It is a nicely written manuscript, easy to follow. However, I have few suggestions/ questions to improve the quality and readability of the article.
- Abstract: any kind of numerical values are unnecessary in the abstract.
- Have you tried NMDAR based anesthesia, if not why only GABA based?
- Does the dose and duration of the anesthesia sufficient to induce synaptogenesis. Need to be mentioned clearly.
- Introduction is too long; it can be short and concise.
- Results: Most of the results written in the text are duplicated in the figures. Values written in the text can be removed to improve the readability. However, these values can be added to the graphs if needed.
- Different color for different treatments would improve the figures.
- For comparisons of the multiple groups, One-Way-ANOVA, which is a more powerful text, should be used.
- All graphs should be combined into one or max two figures. The figure can have sub-figures.
- Tables: There is confusion of “coma (,)” and “decimal (.)” in the values. As a standard usages decimals are used as “.” Not coma.
- I think the table-1 and table-2 are exchanged; the description looks mismatched.
- Why term “body mass” is used instead of body weight, explain. If no specific purpose, then “body weight” is standard usages.
Author Response
Reviewer 2
Dear reviewer,
thank you for Your comments and suggestions. I have addressed all points highlighted in Your review below and used colour marking in the manuscript to improve readability of the corrections.
Piwowarczyk et al., showed that childhood exposure to propofol and sevoflurane anesthesia did not affect the seizure threshold in adult rats. It is an interesting piece of study having a translational value. It is a nicely written manuscript, easy to follow. However, I have few suggestions/ questions to improve the quality and readability of the article.
- Abstract: any kind of numerical values are unnecessary in the abstract.
Authors’ response: We have removed numerical values from the abstract according to your suggestions.
- Have you tried NMDAR based anesthesia, if not why only GABA based?
Authors’ response: We have not tried to investigate the influence of agents that work via NMDAR antagonism on seizure threshold. We focused on the two most commonly used agents in the pediatric setting: propofol and sevoflurane. Use of NMDAR antagonists during synaptogenesis will be our next focus of research.
- Does the dose and duration of the anesthesia sufficient to induce synaptogenesis. Need to be mentioned clearly.
Authors’ response: The anesthetic protocol regarding propofol anesthesia applied in the presented manuscript was identical as in the study presented by Bercker et al. (2009). The authors of the aforementioned study have demonstrated that anesthesia with a cumulative intraperitoneal dose of 90 mg/kg of propofol caused significantly higher scores of degenerated neurons compared to controls in histological silver nitrate and cupric nitrate staining of rats anesthetized on sixth postnatal day of life. Neurodegenerative changes were mainly present in the thalamus and in subiculus, which is a part of the hippocampus, one of the most epileptogenic regions of the brain. According to the study by Bercker et al. no changes in the CA1 and dentate gyrus were observed after administration of propofol [4,41].
The anesthetic protocol regarding sevoflurane anesthesia in the presented study was based on 6 hour exposure to sevoflurane at the concentrations of 2,5-3,5 vol% on the sixth postnatal day. Using a similar protocol, Tong et al. have proven that 4-hour anesthesia with sevoflurane at the concentration of 3 vol% has caused a significant increase in apoptosis in newly-born granule cells of dentate gyrus in rats 2 h, 24 h, 4 days, and 7 days after exposure in juvenile rats. Denate gurus, which is a region of the hippocampus, is strictly associated with epileptogenesis [13,42].
According to the regulations of the Helsinki Declaration, to minimize the number of animals in the experiments that have already been conducted, we did not repeat histological examination.
- Bercker S, Bert B, Bittigau P i wsp.. Neurodegeneration in newborn rats following propofol and sevoflurane anesthesia. Neurotox Res. 2009; 16: 140-147.
- Lévesque M, Avoli M. The subiculum and its role in focal epileptic disorders. Rev Neurosci. 2020 Nov 30;32(3):249-273. doi: 10.1515/revneuro-2020-0091. PMID: 33661586.
- Tong D, Ma Z, Su P, Wang S, Xu Y, Zhang LM, Wu Z, Liu K, Zhao P. Sevoflurane-Induced Neuroapoptosis in Rat Dentate Gyrus Is Activated by Autophagy Through NF-κB Signaling on the Late-Stage Progenitor Granule Cells. Front Cell Neurosci. 2020 Dec 15;14:590577.
- Dengler CG, Coulter DA. Normal and epilepsy-associated pathologic function of the dentate gyrus. Prog Brain Res. 2016;226:155-178. doi:10.1016/bs.pbr.2016.04.005
- Fang F, Xue Z, Cang J. Sevoflurane exposure in 7-day-old rats affects neurogenesis, neurodegeneration and neurocognitive function. Neurosci Bull. 2012 Oct;28(5):499-508.
- Zhao Y, Chen K, Shen X. Environmental Enrichment Attenuated Sevoflurane-Induced Neurotoxicity through the PPAR-γ Signaling Pathway. Biomed Res Int. 2015;2015:107149.
We have included relevant additional sentences in the introduction, methods section, and discussion.
In the Introduction:
„Prolonged exposure of developing rats to sevoflurane and propofol has been proven to cause neurodegeneration and apoptosis in the brains [4,13].”
In the Methods:
„According to protocol described by Tong et al. (2020) [13] the rats in the sevoflurane group (thirty rats) were placed in a plastic chamber (EZ Anesthesia, Palmer, USA) and exposed to 2.5–3.5% sevoflurane (Abbott Laboratories, Abott Park, IL, USA) for 6 h with oxygen and air mixture (30% oxygen) as a carrier at a gas flow of 2 l/min.”
„According to the protocol described by Bercker et al. (2009) [4] propofol was injected intraperitoneally to thirty rats at 30 mg/kg body weight (Diprivan, AstraZeneca, GB) every 90 min until a cumulative dose of 90 mg/kg was reached. In the control group, 0.9% NaCl was injected intraperitoneally to thirty rats under the same dosing regimen.”
In the Discussion:
In order to test the hypothesis that neuropathological alterations induced by propofol and sevoflurane during synaptogenesis in rats may lead to changes in seizure threshold during adulthood we adopted protocols described by Bercker et al. and Tong et al. [13].
In his original study Bercker et al. has demonstrated that propofol administered to 6-days old rats at the cumulative dose of 90 mg/kg produced profound neurodegenerative effect. The significantly higher scores of degenerated neurons compared to controls were detected using histological silver nitrate and cupric nitrate staining. Moreover, morphological changes were followed by persistent cognitive and learning deficits [4]. Importantly, neurodegeneration was mainly present in the thalamus and in subiculus, which is a part of the hippocampus, one of the most epileptogenic regions of the brain [41]. This protocol was accurately reproduced in our study.
In our study the exposure of rats to sevoflurane was carried out according to Tong et al. [13] with minor modification. In our study rats were younger (D6 vs D21) and time of exposure was longer (6 h vs 4 h). Tong and co-workers performed in-depth morphological analysis of the brain. They have proven that sevoflurane anesthesia in juvenile rats has caused a significant increase in apoptosis in newly-born granule cells of dentate gyrus in rats 2 h, 24 h, 4 days, and 7 days after exposure. Additionally, they demonstrated that exposure to sevoflurane decreased the differentiation of the BrdU-labeled late-stage progenitor granule cells but increased the expression of caspase-3, autophagy, and phosphorylated-P65 in the hippocampus of juvenile rats and resulted in cognitive deficiency [13]. Noteworthy, denate gurus, which is a region of the hippocampus, is strictly associated with epileptogenesis [42].
Furthermore, there are many in vivo and in vitro reports that confirm neurodegeneration after anesthetic regimen similar to the one used in the presented manuscript. According to Fang et al. 4-hour sevoflurane exposure on D7 decreased the number of BrdU-positive cells in the hippocampal dentate gyrus and Ki-67 mRNA, indicating neurogenesis is suppressed. Sevoflurane additionally enhanced Fluoro-Jade staining in the dentate gyrus and induced caspase-3 activation indicative of neuronal cell death until at least 4 days after anesthesia. Spatial reference memory was affected at 6 weeks after sevoflurane administration [43]. Moreover, Zhao et al. has confirmed that 3% sevoflurane administered for 2 hours daily from postnatal day 6 to P8 induced neuronal apoptosis and neurogenesis inhibition through the PPAR? signaling pathway in 6-day-old mice, which impaired learning and memory in young mice [44].”
- Introduction is too long; it can be short and concise.
Authors’ response: We have shortened the introduction according to your suggestion. The paragraph about epilepsy and a few sentences in the paragraph that described anesthetics have been removed. (overall reduction by 112 words.)
- Results: Most of the results written in the text are duplicated in the figures. Values written in the text can be removed to improve the readability. However, these values can be added to the graphs if needed.
Authors’ response: We have removed the duplicated data and numerical values from the text according to your suggestions to improve the readability.
- Different color for different treatments would improve the figures.
Authors’ response: We have used a different color for the three different study groups according to your suggestions.
- For comparisons of the multiple groups, One-Way-ANOVA, which is a more powerful text, should be used.
Authors’ response: Thank you for your remark. We have compared multiple groups regarding body weight with the use of One Way ANOVA instead of comparing only the study group with the control group with the Mann-Whitney test. In regards to the corneal kindling model - due to the fact that this seizure model is based on ordinal values instead of continuous values, we used the Mann-Whitney U Test.
- All graphs should be combined into one or max two figures. The figure can have sub-figures.
Authors’ response: We have changed the number of figures from 5 to 2. The first figure depicts the influence of anesthetics on 4th and 5th grade seizures in the corneal kindling model, and the second figure depicts rats' body weight at D6, D30, and D60 in the four different study groups.
- Tables: There is confusion of “coma (,)” and “decimal (.)” in the values. As a standard usages decimals are used as “.” Not coma.
Authors’ response: We have changed coma (,) into decimal (.) in all the values in the manuscript according to your suggestion.
- I think the table-1 and table-2 are exchanged; the description looks mismatched.
Authors’ response: There was no separation between table 2 and the manuscript paragraph about body weight. We have increased the space between table 2 and the manuscript not to confuse the reader.
- Why term “body mass” is used instead of body weight, explain. If no specific purpose, then “body weight” is standard usages.
Authors’ response: We have changed the term „ body mass” to „body weight” in the manuscript according to your suggestions.

Round 2
Reviewer 1 Report
Thank you for reviewing the manuscript. It's ready to be published in my opinion.